# tRNA Core Hypothesis for the Transition from the RNA World to the Ribonucleoprotein World

**DOI:** 10.3390/life6020015

**Published:** 2016-03-23

**Authors:** Savio T. de Farias, Thais G. Rêgo, Marco V. José

**Affiliations:** 1Laboratório de Genética Evolutiva Paulo Leminsk, Departamento de Biologia Molecular, Universidade Federal da Paraíba, João Pessoa 58051-900, Brazil; 2Departamento de Informática, Universidade Federal da Paraíba, João Pessoa 58051-900, Brazil; thaisgaudencio@gmail.com; 3Theoretical Biology Group, Instituto de Investigaciones Biomédicas, Universidad Nacional Autónoma de México, México 04510, Mexico

**Keywords:** tRNA, origin of life, translation system, ribosome

## Abstract

Herein we present the tRNA core hypothesis, which emphasizes the central role of tRNAs molecules in the origin and evolution of fundamental biological processes. tRNAs gave origin to the first genes (mRNA) and the peptidyl transferase center (rRNA), proto-tRNAs were at the core of a proto-translation system, and the anticodon and operational codes then arose in tRNAs molecules. Metabolic pathways emerged from evolutionary pressures of the decoding systems. The transitions from the RNA world to the ribonucleoprotein world to modern biological systems were driven by three kinds of tRNAs transitions, to wit, tRNAs leading to both mRNA and rRNA.

## 1. Introduction

The formation of Earth was a violent cosmic event (*circa* 4.5 billion years ago), and shortly afterwards (*circa* 3.7 billions of years ago) particular conditions led to the emergence of a special state of matter that we call life. One of the main features of life is its capacity for self-generation, or self-reference, which is the property of biological systems to self-organize and to transmit genetic information with some modifications along generations (Darwinian evolution). At the core of this ability of self-referral, a primeval genetic code emerged for the translation of nucleic acids to proteins. Another vital property is the capacity of replication and chemical conversion of this information. Several experiments that simulate Earth’s primitive conditions have yielded the basic molecular components, which are still essential in the modern forms of life [1,2,3,4]. The discovery of catalytic activity by RNA molecules [5,6] opened a new way of thinking about the origins of biological systems. RNA molecules could simultaneously possess the system’s information and perform activities for their self-replication. This model of the origin of life is called the “RNA World Hypothesis” [7,8,9,10,11,12]. Another hypothesis for the origin of life is based on the abundance of amino acids and the diversity of functions performed by proteins. It is proposed that biological systems began essentially as a protein system which is dubbed the “Protein World” or “Protein First Model” [13,14,15]. A third model for the origin of life is based on the interactions between proteins and nucleic acids (RNA), which are the basis of the functioning of biological systems, and this model is called the “ribonucleoprotein world” [16,17]. Di Giulio proposed an early ribonucleoprotein world, based on the intrinsic relationship between nucleic acids and polypeptides observed in modern coenzymes. In his model, the ribonucleoproteins were RNAs covalently linked to polypeptides, such as peptidyl-tRNAs. The nonribosomal synthesis of peptides was conceived [18]. All these models look for answers to fundamental questions about the origin of life as: How did the first genes emerge? How was the information of these genes decoded? How did the metabolic systems arise?

This article presents a novel hypothesis for the transition from the RNA world to the ribonucleoprotein world, where proto-tRNAs molecules possessing similar folds to those observed in modern tRNAs guided the evolutionary process of the genetic code and the translation, and enabled its fixation.

### The Hypothesis

The most plausible scenario of the origin of life is based on RNA molecules that exhibited simple catalytic functions. The tRNA molecules diversified novel structural conformations by the generation of new strands, and they formed new mini-helixes with catalytic function, *i.e.* ribozymes. With the stabilization of the catalytic reactions, these ribozymes began to participate in the first catalytic cycles. At this stage, the structural information emerged and was involved in the direct replication by complementarity between nucleotides. Amino acids in prebiotic conditions were abundant, but their incorporation to primitive forms of life should have been dependent on the chemical interaction with the ribozymes, which introduced a compositional bias induced by hydropathic correlations between amino acids and RNAs. Hydrophobic (hydrophilic) amino acids interacted with hydrophobic (hydrophilic) anticodons, leading to specific interactions, whereby a coding/decoding system of the biological information emerged [16]. At this point, the transition from the RNA world to a ribonucleoprotein world was not only possible but also favored. At the center of this transition, we have to explain how the translation system was organized. Nowadays the translation system is a complex system that involves a complex group of proteins that include ribosomal proteins, aminoacyl-tRNA synthetases, elongation and termination factors, recycling factory, among other proteins, and three kinds of RNA molecules, to wit, mRNA, rRNA, and tRNA. The mRNA contains the information codified in the DNA to synthesize a protein, the rRNA forms the ribosome where amino acids are joined by peptide bonds, and the tRNAs mediate the transmission of information codified in mRNA in the ribosome to accomplish the protein synthesis. The origin of protein biosynthesis machinery is still fertile ground in evolutionary biology and represents a critical evolutionary transition. Eigen (1971) suggested that RNA molecules had a limited capacity for replication without loss of information in an environment free of enzymes [19]. This phenomenon is known as the error catastrophe. The size of the nucleotide chain where biological information began to be conserved was present in relatively few nucleotides, around 70 nucleotides or less, like those of modern tRNAs [20]. It is possible that, in the beginning, tRNA-precursors were smaller than the modern tRNA. These initial molecules were similar to the building blocks of a proto-ribosome and functioned as a bonding apparatus in the RNA world before making peptide bonds [21,22,23,24]. These molecules had the ability to participate in the formation of new functional structures by joining two or more RNA molecules. Then, it was possible to increase the information stored in the RNA molecules, which allowed the appearance of the first genes. Thus, many combinations of proto-tRNAs worked as ribozymes, and their polymeric products gave origin to the peptidyl transferase center (PTC), a catalytic center capable of carrying out peptide bonds [21,25,26,27].

Eigen and Winkler-Oswatisch suggested an Ur-gene made of tRNA [28]. Bloch *et al.*, comparing the tRNAs and 16S ribosomal RNA chain, suggested a common origin for these molecules [29,30]. Agmon showed that the modern tRNAs were built by the same building blocks as the modern, fully conserved PTC [23]. Tamura further analyzed the topological organization of the PTC of the large subunit of the ribosome and suggested that this region presents topological similarities with concatenated tRNAs [27]. Root-Bernstein and Root-Bernstein analyzed the origin of PTC and described its origin as a collection of tRNAs that could function as a primitive and functional genome [31]. Di Giulio suggested that the tRNAs molecules have a non-monophyletic origin and that, in the emergence of the biological system, the primitive genome could have been fragmented, and the trans-splicing events thus may have had an important function in the origin of the genes [32].

The hypothesis here proposed places the tRNA at the core of the origin of translation, *i.e.*, tRNA molecules played a central role in the organization of the first codified biological system, established the information storage system, and participated in coding and decoding this information. They were the protagonists of the translation system via chemical interactions with amino acids, which are inherent to the transition between the RNA world to a ribonucleoprotein world.

The present hypothesis contends that the tRNA-like molecules or proto-tRNAs (the 2D-cloverleaf tRNA canonical structure), through structural modification and concatenation, gave origin to the early genes, as well as to the PTC. In addition, we suggest that metabolic pathways resulted from evolutionary pressures of the decoding system. Hence, the first synthesized peptides by a proto-translation system could have been selected for binding and stabilization of the RNAs that acted as ribozymes and proto-genes [33]. In this manner, concatenated RNAs, which acted as a primitive genome, increased the amount of stored information, and, due to the increased size of RNAs strands, the new peptides emerged in a nascent ribonucleoprotein world. Thus, via positive feedback between peptides and RNA, the biological system developed robustness that was crucial for the establishment of the metabolic and informational processes, as we observe in a modern biological system. Figure 1 summarizes the functions of the tRNA-like molecules at the transition from the RNA world to the ribonucleoprotein world.

When proto-tRNAS joined in concatemers, they acquired new structural conformations, similar to the folds of both the proto-mRNA and proto-rRNA, which by interaction, were able to initiate translations into proteins. At this stage, positive feedback between proteins and RNAs was essential for further evolution.

## 2. Experimental Section

### tRNA and the First Genes

The tRNA sequences were obtained from the tRNA database (http://trnadb.bioinf.uni-leipzig.de) and correspond to 361 organisms distributed in the three domains of life. The RNY code was proposed as a primeval genetic code by Eigen and Schuster (1978). This code displays the property of base complementarity with respect to plus and minus strands, and it minimizes errors in the replication and translation processes [34]. Hence, we reconstructed the ancestor sequences for each type of tRNA with RNY anticodons (Appendix A). The total of 600 sequences for ^Ala^tRNA, 313 for ^Asn^tRNA, 267 for ^Asp^tRNA, 608 for ^Gly^tRNA, 326 for ^Ile^tRNA, 855 for ^Ser^tRNA, 634 for ^Thr^tRNA, and 538 for ^Val^tRNA were analyzed. Model Tests was performed for each group of tRNAs to choose the best evolutionary model, which pointed to Kimura 2 parameters. To reconstruct the ancestral sequences, a phylogenetic tree was generated using the maximum-likelihood method. In order to achieve statistical significance, bootstrapping with 1000 replicates was carried out. These evaluations were performed with the MEGA5 program [35]. From the ancestor sequences for each tRNAs with anticodons of the RNY type (Ile, Asn, Vau, Asp, Gly, Ser, Thr, and Ala), an initial set of tRNAs that composed the progenotes was proposed [34]. Sequences were constructed from all possible combinations (3 × 3) of those elements that take place without the presence of the same tRNA in two positions in the final sequence. A search for similar proteins from the combined ancestral tRNA sequences in NCBI with the BLASTX algorithm [36] in the database UniProtKB/Swiss-Prot (swissprot) was performed. All sequences that had similarities in the database were analyzed (Appendix A) and ranked according to their functional category with the support of the phylogenetic classification of proteins tool encoded in complete genomes by COG (Clusters of Orthologous Groups of proteins) [37].

## 3. Results and Discussion

### 3.1. tRNAs and the Origin of Genes

In this hypothesis, the first genes derived from tRNA by structural changes (tRNA-like-mRNA structure) enabled other tRNAs (cloverleaf tRNA canonical structure) to bind this sequence by the loop of the proto-anticodon. Szathmáry proposed that, in the origin of the biological systems, amino acids worked as cofactors binding to ribozymes that had the primordial function of aminoacylation and, from this interaction, the biological relation between amino acids and specific oligonucleotides emerged. Thereby, this interaction was important for the emergence of the genetic code [39]. Da Silva suggested that the interaction between amino acids and nucleic acids, as riboswitches, was important for the stabilization of the RNA structure and relevant for its catalytic activity as a ribozyme [40]. Di Giulio proposed that the primordial mRNA originated by interactions of peptidated-RNAs (aminoacylated), which had the aspect of hairpin-like structures [41]. In the present proposal, amino acids and small peptides worked like riboswitches or cofactors to stabilize the alternative conformations of the proto-tRNAs. Thus, the binding of two or more tRNAs, which showed distinct structural organization (tRNA-like mRNA and cloverleaf tRNA canonical structure), allowed the stability of these molecules to increase (Figure 2).

The tRNA molecules with the proto-tRNA (anticodon stem/loop) could interact with the amino acids (as cofactor or riboswitches), which were present in the prebiotic environment; hence, the binding between tRNA and amino acids was established. The tRNAs binding to amino acids could interact with other tRNAs in open conformation (tRNA-like mRNA structure). This interaction stabilized the complex cloverleaf tRNA canonical structure-tRNA-like mRNA structure. In addition, this interaction allowed the assembly of the binary complex proto-tRNA/mRNA, to be closer to the tRNAs charged with amino acids, which, after their stabilization, enabled the interaction with the proto-rRNA (PTC), forming the ternary complex proto-tRNA/mRNA/rRNA (PTC) and thus stabilizing the primitive protein synthesis system. Figure 2 portrays the mechanism of interaction between tRNAs that gave rise to the primitive protein synthesis system. It is suggested that the first peptides that were fixed in the biological system had the ability to bind and stabilize tRNAs in different structural conformations, which in turn increased the stability and efficiency of the process, thereby establishing the first positive feedback between peptides and nucleic acids [42,43,44]. With an increased stability of the primitive protein synthesis system, its efficiency increased, and the rate of consumption of amino acids involved in proto-proteins increased. At this stage, the biological system had a new selective pressure because the prebiotic substrates decreased, and the replacement of essential components for maintenance of the system was required (Figure 3).

Thus, the primordial metabolic pathways of amino acid synthesis started its organization, as well as the primitive metabolic pathways of nucleotides catalyzed by enzymes. Keller *et al.* demonstrated that the main reactions of glycolysis and pentose phosphate pathways were plausible in a prebiotic condition [45]. In modern metabolic pathways, those of the glycolysis and pentose phosphate act as a distribution center for precursors of amino acids and nucleotides biosynthesis pathways. Thus, these primordial routes that emerged in an environment without proteins were gradually replaced by primordial enzyme-catalyzed versions after the emergence of the primordial translation system of peptides, which increased the efficiency of this synthesis process, and amino acids were supplied for the translation system in formation. Thus, the translation system became the attractor, the state where the system channeled the production of amino acids, and therefore was the first selective force to organize the metabolic pathways that we observe in modern biological systems.

Table 1 shows the results obtained by the translation of ancestral tRNAs, indicating which motifs had similarities with current proteins. We observe that most of the proteins that had similarities with proto-genes, constructed with ancestral sequences of tRNAs, are involved in the translation process. This implies that they have the capability of binding to RNA. We also note a similarity with the enzyme RNA-polymerase directed by RNA, which replicates RNA molecules. Another fact that stands out is the similarity of the proto-genes with proteins involved in the metabolism of molecules with three carbon, which are present at the core of the glycolytic pathway, which turns out to be a distribution center of molecules to various pathways in modern cells. Among these pathways, we have amino acid synthesis, nucleotide synthesis, lipid synthesis, and energy generation.

Conclusions are consistent with the ones obtained by Delaye *et al.*, who reconstructed the probable metabolic pathways of LUCA via methods based in homology [46]. In this work, a likely proteome for this biological entity is suggested. Thus, the results suggest that the tRNAs may have originated the early genes, which encoded proteins involved in the binding to RNA, and proteins involved in the pathways for the replacement of prebiotic components in peptides synthesis. The results also suggest that the system of encoding/decoding of biological information arose early in the organization of biological systems, where the same tRNAs participated in the information storage. The incorporation of new metabolic pathways may have been selected from their chemical collaboration between the replacement of organic compounds and the process of peptide synthesis. In this context, the modern biological system was initially established as a ribonucleoprotein system, where the nascent proteins could bind to tRNAs and stabilize them. This stabilization enhanced the fidelity of peptides synthesis, which conferred strength to the nascent biological system.

### 3.2. tRNAs and the Origin of Ribosomes

The emergence of the first genes, as well as the need to decode the information contained in these genes, must have been a major challenge in the origin of biological systems. Bloch *et al.* suggested, when comparing the sequences of tRNAs and 16S ribosomal RNA, that these molecules had a common origin [29,30]. Tamura, in a comparative structural study between the PTC and tRNAs, suggested that the former arose by the concatenation of the latter [27]. Davidovich *et al.* analyzed the possibility of small RNAs to acquire a similar structure to the PTC and noted that RNA with structural organization of the type stem-elbow-stem, when joined, had comparable structural organization with the PTC of the ribosome [47]. Root-Bernstein and Root-Bernstein suggested that the ribosome contained information of the tRNAs that functioned as the primordial genome [31]. In the search for evidence that might suggest an origin of the PTC from tRNA-like molecules, ancestral sequences of tRNAs were obtained [38]. From individual alignments of tRNAs with the PTC of *Thermus thermophilus*, it was possible to construct a concatamer of tRNAs and compare its similarity to the PTC as a whole. The ancestral sequences of tRNAs used for the construction of the concatamer and its comparison with the PTC were ^Leu^tRNA-^Ser^tRNA-^His^tRNA-^Pro^tRNA-^Tyr^tRNA-^Phe^-tRNA-^Gln^tRNA-^Gly^tRNA-^Lys^tRNA. The alignment showed an identity of 50.53% between the sequence constructed with ancestral tRNAs and the PTC from *T. thermophiles*. We have previously published the results of an alignment of tRNA ancestral sequences with the PTC in [38].

These findings, along with the results of other groups [26,30], suggest that the tRNA-like molecules originated the catalytic portion of the ribosome and assisted in the decoding of the information contained in the first genes that in turn could have originated from the tRNA molecules [38]. Thus, our hypothesis provides novel insight on the origin of the ribonucleoprotein world, where the tRNAs originated the peptidyl transferase center and the first genes, important elements in the origin of the translation system.

## 4. Conclusions

The present hypothesis suggests that tRNA molecules organized the essence of the biological system, giving rise to the first genes and the catalytic center of the ribosome, the PTC, thereby orchestrating the origin of the primitive protein synthesis system. The first synthesized proteins or peptides could bind to the RNA apparatus that produced them and stabilize the other tRNA-like molecules in its specific functional conformations, as well as act as a primitive genome, thus increasing the robustness of the biological system in formation. The data obtained from the analysis of the reconstruction of the ancestral sequences of the tRNAs showed evidence of the cardinal involvement of the tRNAs in the origin of the early genes. These genes encoded proteins which were involved in protein synthesis and the metabolism of molecules with 3 carbons of the glycolytic pathway. The latter is part of the distribution and supply of compounds for various other metabolic pathways. The data also show that tRNA-like molecules may have led to the assembly of the catalytic center of the ribosome. From the data, we suggest that the tRNAs were the key molecules in the organization and evolution of the biological systems, initiating the feedback relationships between proteins and nucleic acids observed today in modern organisms. Thus, we propose a model about the transitions from the RNA world to ribonucleoprotein world to the origin of the modern biological systems, where the system encoding/decoding emerged as a natural pressure for the maintenance of the primitive translation system.

## Figures and Tables

**Figure 1 life-06-00015-f001:**
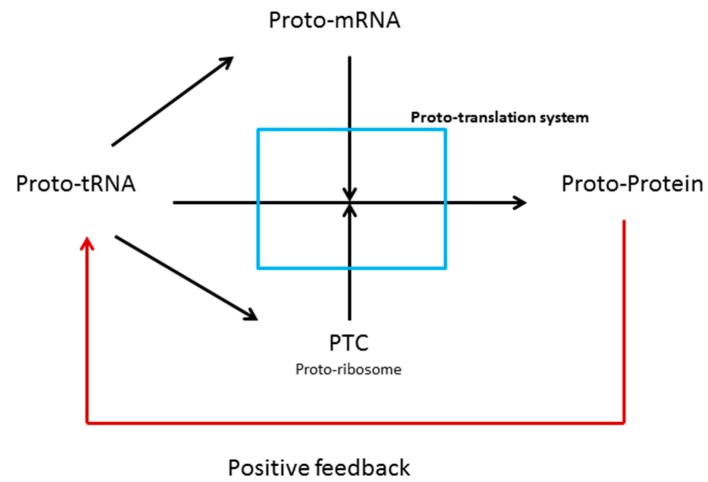
Schematic model of the origin of the translation system originated from proto-tRNAs.

**Figure 2 life-06-00015-f002:**
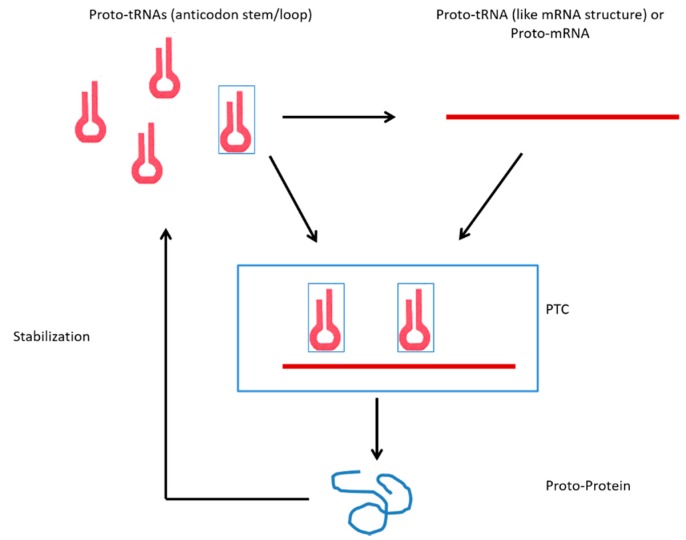
Schematic model of the origin of the translation system by the change of the tertiary structure of proto-tRNAs (anticodon stem/loop) and the mechanisms of interactions between the alternative structural states.

**Figure 3 life-06-00015-f003:**
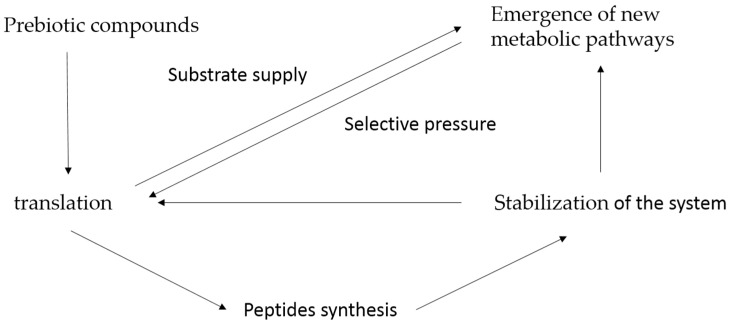
Sequential stages in the emergence of the translation system and the initial selective pressure that originated the first metabolic pathways.

**Table 1 life-06-00015-t001:** Enzymes and processes that matched between ancestor tRNA and the modern proteins. The original BLAST results can be found in the Appendix A.

Protein	Process	Protein	Process
Elongation factor 1–alfa RNA transport	Translation	Putative ribose/galactose/methyl galactose import	Glycolysis/glycogenesis
Elongation factor 4–mRNA translation assisting	Translation	Glycerate kinase	Glycolysis/glycogenesis
tRNA uridine 5-carboxymethylaminomethyl modification	Translation	Triosephosphate isomerase	Glycolysis/glycogenesis
60S ribosomal protein L3	Translation	Beta-glucosidase A	Glycolysis/glycogenesis
60S ribosomal protein L7a	Translation	Glucose-6-phosphate-1-dehydrogenase	Glycolysis/glycogenesis
60S ribosomal protein L27a-1	Translation	Glucose 6 phosphate isomerase	Glycolysis/glycogenesis
60S ribosomal protein L27a-3	Translation	Phosphoglycerate kinase	Glycolysis/glycogenesis
60S ribosomal protein L27a-2/4	Translation	Glycerol-3-phosphate dehydrogenase	Glycolysis/glycogenesis
50S ribosomal protein L10	Translation	Transketolase	Glycolysis/glycogenesis
Methionyl–tRNA formyl transferase	Translation	α-galactosidase	Glycolysis/glycogenesis
Lysys-tRNA synthetase	Translation	Diaminopimelate epimerase	Amino acids pathways
Asparaginyl-tRNA synthetase	Translation	l-asparaginase	Amino acids pathways
Glutamyl-tRNA synthetase	Translation	ATP phosphoribosyl transferase	Amino acids pathways
Leucyl-tRNA synthetase	Translation	Histidinol phosphate aminotransfarase	Amino acids pathways
Valyl-tRNA synthetase	Translation	4-aminobutyrate aminotransferase	Amino acids pathways
Phenylalanine-tRNA synthetase	Translation	Ornithine carboxylase antizyme	Amino acids pathways
RNA methyltransferase—tRNA modification	Translation	*N*-acetyl-γ-glutamyl-phosphate redutase	Amino acids pathways
DNA-direct RNA polymerase or RNA-direct RNA polymerase	Transcription	Homoserine kinase	Amino acids pathways
Thymidylate kinase	Nucleotides pathways	Aromatic amino acid aminotransferase	Amino acids pathways
Cytidine deaminase	Nucleotides pathways	Ornithine carbomyltransferase	Amino acids pathways
Uridylate kinase	Nucleotides pathways	Tryptophan synthase α-chain	Amino acids pathways
Orotidine 5-phosphate decarboxilase	Nucleotides pathways	Fatty acid synthase	Lipids pathways
Dihydroorotate dehydrogenase	Nucleotides pathways	CoA mutase	Lipids pathways
Phosphoribosyl formyl glycinamidine cyclo-ligase	Nucleotides pathways	Phosphate acyltransferase	Lipids pathways
Phosphoribosyl glycinamidine synthase	Nucleotides pathways	Lycopene cyclase	Lipids pathways
		3-β-hydroxisteroid dehydrogenase	Lipids pathways

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
