# Peer review of "tRNA Core Hypothesis for the Transition from the RNA World to the Ribonucleoprotein World"

_life, 2016, doi:10.3390/life6020015_

Round 1

Reviewer 1 Report

I am Massimo Di Giulio.

I sincerely think that authors have to talk about and to quote the following my papers to improve their manuscript. If this revision will be performed in an appropriate way then the manuscript will improve and might be accepted.

J Mol Evol. 1997 Dec;45(6):571-8.

On the RNA world: evidence in favor of an early ribonucleopeptide world.

J Mol Evol. 2015 Aug;81(1-2):10-7. 

A Model for the Origin of the First mRNAs.

J Theor Biol. 2006 Jun 7;240(3):343-52. 

The non-monophyletic origin of the tRNA molecule and the origin of genes only after the evolutionary stage of the last universal common ancestor (LUCA).

J Theor Biol. 1992 Nov 21;159(2):199-214.

On the origin of the transfer RNA molecule.

Reviewer 2 Report

I found the English language and style at times made it difficult to understand the authors' meaning, for example in lines 96-98: "Amino acids in prebiotic conditions were abundant, but their incorporation in a nascent biologic system should have been dependent of the chemical interaction with the ribozymes...", with the phrase presumably meant to read either ''...dependent on the chemical interaction..." or "...independent of the chemical interaction...".

There were also a few references with no explanation, such as (in line 71): "...self-generation or self-reference..." The inheritance of information with modification is normally refered to as Darwinian evolution, while the term 'self-reference' will be obscure to most readers, referring as it does to work by Romeu Guimaraes, although his 2011 paper [17] which has this term in its title isn't cited until line 83.

In section 2.1 "tRNA and first gene analysis" of the Materials and methods section, I would have found it helpful if the ancestral tRNA sequences generated were included, perhaps as supplementary material. There is also reference to using only tRNAs with RNY anticodons, but the reasons for doing this are not discussed. There needs to be some discussion of this point here, with references to some of the earlier publications on this e.g. Eigen and Schuster etc. I am not sure that Maximum likelihood is the best method to use to generate such sequences. It would be a good idea for the authors to consult someone in the field with expertise in tRNA phylogenetics, such as Professor Rob Knight from the University of Colorado Boulder, to get advice on this aspect. I was also surprised at the lack of detail regarding the analysis comparing potential tRNA coding sequences and protein sequences, where there is no detail given as to the results of the homology search other than a list of proteins and their functions (in Table 1). I would like to see the homologous sequences for myself so that I could make a judgement on their significance. I am unsure too of the way in which the nine ancestral tRNA sequences were ordered to do the Blast search, as the wording in this section is somewhat clumsy. I find the phrase (lines 166-167) "All analyzed sequences were similar regardless of the level of similarity with the query sequence..." difficult to understand. On a minor note, 'glycolysis' is misspelt a number of times in the last column of Table 1.

Other points:

line 78: the model is known as the RNA world hypothesis – not the "ancient RNA world".

line 81: I'm not sure that Gustavo Caetano-Anolles [14,15] would consider his model 'the protein world'. There is a model known (colloquially at least) as 'proteins first', but I'm not sure that Gustavo completely subscribes to this. The comparison of the three models is fairly superficial: "these models have their own views", and then the authors go on to describe a transition from the RNA world to the ribonucleoprotein world – what has happened to the proteins first model? Is there something the authors don't like about it. If so, they need to state this.

Line 98: the phrase "that introduced a compositional bias induced by hydropathic correlations between amino acids and RNA" needs some kind of explanation and possibly a reference or two.

Line 103: the modern translation system involves more than "several kinds" of proteins.

Lines 110-111: "The size of the nucleotide chain...was present in relatively few nucleotides" is not well expressed.

Line 117: "...many combinations of proto-RNAs worked as ribozymes..." Is there any evidence from modern biology of tRNAs functioning as ribozymes?

Line 121: you write that "tRNA molecules orchestrated the organization of the first codified biological system..." How do the authors propose they did this, or have I missed something?

Lines 125-126 (also in lines 266-268): In [28,29], Bloch et al. compare tRNA sequences with those of the small ribosomal subunit RNA (e.g. the 16S rRNA in E. coli), not the 5S rRNA.

Line 148: the authors state "...the folds of the proto-tRNA were similar to the folds of both the proto-mRNA and proto-rRNA..." Wouldn't folds or secondary structure be detrimental to the function of a proto-mRNA, in that it needs to be 'read' and therefore mainly in a single-stranded conformation? On th contrary, isn't the evolution of proto-tRNA into proto-mRNA more likely to have occurred through the loss of secondary structure, and – if this is so – how do the authors propose this would have happened?

Line 155: as mentioned above, the concept of tRNAs with RNY anticodons is introduced with no explanation; the significance of this sequence will be lost on many readers. You need to discuss the ideas behind this model and refer to the main papers e.g. by Eigen and Schuster etc.

Lines 182-185: some explanation of why you were interested in the frequencies of dinucleotides might be helpful here.

Lines 193-194: some explanation of Szathmary's proposal that amino acids acting as cofactors were important in the emergence of the genetic code would be useful here.

Figure 2: what you have labeled as "Proto-tRNAs (canonical structure)" in the figure look like hairpin loops – are these supposed to represent the anticodon stem/loop? If so, this needs to be described better in the figure and in the figure legend. 

Lines 205-206: How do the authors propose that the tRNAs molecules interacted with the amino acids as cofactors or – especially – riboswitches? Is there any evidence of tRNAs binding small molecules? The only example I can think of is the binding by tRNA of Pb2+ and its subsequent cleavage (Brown et al. (1983) Nature 303:543-6).

Lines 221-222: this sentence does make sense as it is written.

Lines 226-227: How do the authors propose that "the primordial metabolic pathways of amino acid synthesis started its organization"?

Lines 230-232: "Thus, these primordial routes that emerged in an environment without proteins, were gradually replaced by primordial enzyme(s)-catalyzed versions, after the emergence of the primordial translation system of peptides..." But if the authors are proposing the evolution of a ribonucleoprotein world from an earlier RNA world (as stated in the introduction), wouldn't there have been an intermediate stage when the metabolic etc reactions were catalyzed by ribozymes?

Line 239: By the phrase proteins "involved in the translation process", are the authors meaning ribosomal proteins?

Line 251: "conclusions are consistent with" is better.

Lines 261-263: the same thing as before regarding use of the phrase "self-reference"; and "could persist for the rest of the history of life on Earth" is probably unnecessary.

Lines 266-267: misquote of Bloch et al. again.

Lines 269-272: Sadly, Davidovich et al. haven't yet managed to show that their self-folded SES dimers are able to catalyze peptide (or any other form of) synthesis.

Line 287: "...the results of other groups..." What other groups? This comment needs elaboration and citations.

Linew 291-292: "...where (the) tRNA commanded the organization, and consequently the whole development of biological systems". This is a very big claim, and I am not sure that your results support this.

Line 309: this line needs rewording.

Line 311: should be Dr Ada Yonath.

Round 2

Reviewer 2 Report

Firstly, I would like to thank the authors for bringining to my attention a number of recent references to the use of the Maximum likelihood method to reconstruct ancestral sequences. Also, for pointing out the statement by Caetano-Anolles etc al. in [14,15] in support of a proteins-first view - this is very clear!

However, I still have a number of problems with this manuscript, the main one being the re-use of results previously published as Figure 1 in [38] (Farias et al. 2014) without full acknowledgement of this fact.

1. It states in section 2.2 of the Experimental Section "tRNA and the PTC analysis", line 160: "We followed the same methodology by Farias et al., 2014 [38]". As these results (and their methods) have previously been published, this section is unnecessary and should not be included.

2. In section 3.2 "tRNAs and the origin of ribosomes", line 274 states "...we used the ancestral sequences of tRNAs and we aligned them with the PTC..." Instead, it should state, "We have previously published the results of an alignment of tRNA ancestral sequences with the PTC in [38]."

3. In the legend to Figure 4, line 284 states "Adapted from Farias et al., 2014 [37]". This reference is incorrect (it should be [38]), and should read something like "Figure taken from [38], used with permission" (if indeed permission has been obtained from the editors of the previous publication). The figure is actually exactly the same as that published in [38], except that the authors have changed the left-hand side annotation and some of the sequence numbering on the right-hand side. As the sequence alignments appear to be identical between the two figures, this raises the question of which numbering is the correct one.

As this section is is a review of previous work, it only needs to be referred to briefly (and possibly doesn't even require a separate section).

While I appreciate the authors have now included the reults of the Blast search used to derive Table 1 in the supplementary information section of the manuscript, I believe the importance of this data is such that the sequences should be shown as part of Table 1 within the body of the paper. If this makes the resultant table too large, I suggest reducing the number of protein 'hits' to those with the highest homology, either that or only showing a few of the representative proteins from the different processes (e.g. translation etc). There is plenty of room to do this, as the table as it currently stands is doubled up, having two columns for proteins and two for process, which is unnecessary. The table also needs to rank the proteins in order of homology. The remaining data not included in Table 1 can be shown in the supplementary information, if desired.

A minor point: in the legend to Table 1, lines 236-7, it states that this table "[indicates] which motifs had similarities with current proteins". Where in the table are these motifs shown?

I am not sure how the authors have compiled the Blast search results in the Supplementary Information section, as unfortunately I was unable to open the supplementary file as it is in a format that my computer does not recognize.

Finally, while I appreciate English is probably not the first language of the authors, I would strongly recommend employing a professional (service) to edit the manuscript for readibility, as there are still major difficulties with the wording and grammar.

Round 3

Reviewer 2 Report

The manuscript has been much improved, although I still have some concerns about the standard of English language used. My only other query relates to Supplement 1, which contains the ancestral sequence of cognate tRNAs. The way these sequences are presented, it is not clear to me how they relate to the full tRNA sequences. For example, the length of the sequences vary between 25 and 57 nucleotides, and the ancestral serine sequence (25 nucleotides) does not appear to contain any serine anticodon.